# Fracture of Epoxy Matrixes Modified with Thermo-Plastic Polymers and Winding Glass Fibers Reinforced Plastics on Their Base under Low-Velocity Impact Condition

**DOI:** 10.3390/polym15132958

**Published:** 2023-07-05

**Authors:** Ilya V. Tretyakov, Tuyara V. Petrova, Aleksey V. Kireynov, Roman A. Korokhin, Elena O. Platonova, Olga V. Alexeeva, Yulia A. Gorbatkina, Vitaliy I. Solodilov, Gleb Yu. Yurkov, Alexander Al. Berlin

**Affiliations:** 1N.N. Semenov Federal Research Center of Chemical Physics, Russian Academy of Sciences, 119991 Moscow, Russia; tuyara.2312@mail.ru (T.V.P.); alexkireinov@gmail.com (A.V.K.); korohinra@gmail.com (R.A.K.); e-o-platonova@yandex.ru (E.O.P.); 1.vitalyo@gmail.com (Y.A.G.); berlin@chph.ras.ru (A.A.B.); 2A.N. Nesmeyanov Institute of Organoelement Compounds, 119334 Moscow, Russia; 3N.M. Emanuel Institute of Biochemical Physics, Russian Academy of Sciences, 119334 Moscow, Russia; alexol@yandex.ru

**Keywords:** epoxy resin, GFRP, thermoplastic polymers, fracture, low-velocity impact conditions

## Abstract

The work is aimed at studying the impact resistance of epoxy oligomer matrices (EO) modified with polysulfone (PSU) or polyethersulfone (PES) and glass fibers reinforced plastics (GFRP) based on them under low-velocity impact conditions. The concentration dependences of strength and fracture energy of modified matrices and GFRP were determined. It has been determined that the type of concentration curves of the fracture energy of GFRP depends on the concentration and type of the modifying polymer. It is shown that strength σ and fracture energy E_M_ of thermoplastic-modified epoxy matrices change little in the concentration range from 0 to 15 wt.%. However, even with the introduction of 20 wt.% PSU into EO, the strength increases from 164 MPa to 200 MPa, and the fracture energy from 32 kJ/m^2^ to 39 kJ/m^2^. The effect of increasing the strength and fracture energy of modified matrices is retained in GFRP. The maximum increase in shear strength (from 72 MPa to 87 MPa) is observed for GFRP based on the EO + 15 wt.% PSU matrix. For GFRP based on EO + 20 wt.% PES, the shear strength is reduced to 69 MPa. The opposite effect is observed for the EO + 20 wt.% PES matrix, where the strength value decreases from 164 MPa to 75 MPa, and the energy decreases from 32 kJ/m^2^ to 10 kJ/m^2^. The reference value for the fracture energy of GFRP 615 is 741 kJ/m^2^. The maximum fracture energy for GFRP is based on EO + 20 wt.% PSU increases to 832 kJ/m^2^ for GFRP based on EO + 20 wt.% PES—up to 950 kJ/m^2^. The study of the morphology of the fracture surfaces of matrices and GFRP confirmed the dependence of impact characteristics on the microstructure of the modified matrices and the degree of involvement in the process of crack formation. The greatest effect is achieved for matrices with a phase structure “thermoplastic matrix-epoxy dispersion.” Correlations between the fracture energy and strength of EO + PES matrices and GFRP have been established.

## 1. Introduction

Polymer composite materials are widely used as structural materials for the manufacture of products for various purposes [1,2,3,4]. High elastic-strength properties of composite materials are provided not only by fibers, but also by the matrix under the condition of high adhesive strength of the polymer-fiber interface. Given the fact that the strength characteristics of polymer matrices are much lower than those of reinforcing fibers, the fracture of reinforced plastics will begin with the growth and accumulation of microcracks in the polymer matrix [5]. Moreover, the strength properties of the matrix will be decisive in the fracture of reinforced plastics from shear or delamination, that is, in cases where the propagation of cracks during the fracture of the material proceeds mainly along the matrix.

Epoxy matrices meet most of the requirements for a polymer matrix. This is primarily due to their high elastic-strength characteristics, adhesion to most high-strength fibers [6], and resistance to chemical [7] and physical influences [8]. At the same time, epoxy matrices are characterized by low crack resistance and impact resistance [9]. As a rule, it is possible to increase crack and impact resistance by modifying epoxy matrices with particles of different nature [10,11,12,13,14,15,16,17,18,19], active diluents [20,21,22,23], rubbers [24,25,26,27], hyperbranched oligomers and polymers [28,29,30,31,32], thermoplastics [9,16,33,34,35,36,37,38,39,40,41,42,43,44,45,46,47,48,49,50].

The use of carbon nanoparticles for epoxy matrices as modifiers often makes it possible to increase their crack resistance. For example, in [11,39,40,41,42,43,44,45,46,47,48,49,50], an increase in the crack resistance of epoxy nanocomposites containing particles of different natures by about 20–70% is noted. An increase in crack resistance is usually associated with an increase in the crack path when rounding particles. The absence of plastic deformations in the matrix does not allow for obtaining an even greater effect from the introduction of particles.

Modification of epoxy matrices with rubbers [9,28,29,30,39,40,41,42,43,44,45,46,47,48,49,50] and active diluents [12,13,31,39,40,41,42,43,44,45,46,47,48,49,50] makes it possible to achieve a greater increase in crack resistance than when particles are added. The heat resistance of matrices with such modification decreases.

Hyperbranched polymers and oligomers can also increase the crack resistance of epoxy matrices [28,30]. The values of crack resistance G_IR_ for the materials studied in [30] increase from 0.12 to 1.5 kJ/m^2^; in [7], the crack resistance K_IC_ increases by 175% relative to the unmodified matrix. 

The highest values of crack resistance (up to 2 kJ/m^2^) can be obtained for epoxy matrices modified with rigid chain polymers [16,21,22,23,24,25,26,27,33,35,37,39,40,41,42,43,44,45,46,47,48,49,50]. The high crack resistance of such materials is associated with the heterogeneous structure of the modified matrices, which arises during the phase decomposition of the initial homogeneous epoxy-polymer mixtures [36,37,39]. In this system, the growing crack not only bends around the more plastic phases of the thermoplastic, but also significantly slows down, passing through it. By reinforcing such modified matrices with high-strength fibers, it is possible to maintain the effect of increasing crack resistance [43,51,52].

The literature data described above were obtained under quasi-static loading conditions. In the event of an impact, the fracture of the sample can occur in a few milliseconds. The effect of modifying epoxy matrices, in this case, may differ significantly from that discussed above.

In [11], an increase in the impact resistance of epoxy matrices modified with graphene by 20–30% is noted. The introduction of SiO_2_ particles coated with a poly(butyl acrylate) shell into an epoxy matrix increases the Izod impact resistance from 0.8 to 1.7 kJ/m^2^ [13]. Unfortunately, the authors focused on the study of the morphology of the boundary layers of SiO_2_ particles and did not consider the mechanisms of hardening of such materials. A similar result with the introduction of SiO_2_ into the epoxy matrix was obtained in [14]. The impact resistance with the introduction of 9 wt.% SiO_2_ increases from 15 to 40 kJ/m^2^. The authors explain the increase in impact resistance by the branching of the crack during the fracture of the material. The effect of introducing particles into matrices is also preserved for reinforced plastics [10]. There is an increase in the impact properties of reinforced plastics from 13 to 45% relative to unmodified systems, depending on their type.

A more significant increase in the impact resistance of polymer matrices is noted when thermoplastic polymers are used as modifiers. For example, in [34], the introduction of 10 wt.% polyethernitrile ketone into an epoxy matrix almost doubles the impact resistance of hybrid matrices (from 18 kJ/m^2^ to 37 kJ/m^2^). Modification with polysulfone also increases the impact resistance of epoxy matrices [39]. At a modifier concentration of 15 wt.% in the epoxy matrix, the impact resistance increases from 23 to 38 J/m^2^. The increase in impact resistance is associated with the heterogeneous structure of the matrix. However, in this work, it is not indicated which fracture surface was studied. A study of the morphology of fracture surfaces of matrices showed that unmodified matrix fractures brittlely. The fracture surface relief is almost smooth. In modified matrices, the fracture surface has multiple cracks and a developed surface relief. A significant increase in the impact resistance of modified matrices is explained by multiple cracking.

An analysis of the literature data revealed the almost absence of comprehensive studies of the impact resistance of matrices and composites. The mechanisms of formation and propagation of cracks are practically not described. This especially applies to hybrid polymer matrices with a complex phase composition and plastics reinforced with continuous fibers.

This work is a continuation of the research work begun earlier [50]. This study is devoted to the determination and analysis of the mechanisms of fracture of thermoplastic-modified epoxy matrices and fiberglasses based on them under low-speed loading conditions. Two thermoplastics, polysulfone, and polyethersulfone, were used as modifiers. The use of polyethersulfone forms polymer mixtures that are described by a phase diagram with a lower critical mixing point, polysulfone—an upper one. The final structure of heterogeneous matrices will be achieved at significantly different temperatures and concentrations [38]. Evaluation of differences between two polymer systems with different mechanisms of phase structure formation will create new opportunities for further study of the micromechanics of fracture of similar heterogeneous matrices. Comparison of the phase structures of matrices and reinforced plastics will make it possible to reveal the features of crack formation for matrices located in cramped conditions and to determine the optimal phase organization of these systems. Optimization of the phase structure of matrices will improve the strength and performance properties of reinforced plastics and increase their life cycle.

## 2. Materials and Methods

### 2.1. Materials and Polymers Mixtures Preparation

Two types of epoxy binder were used in the work. The first type of epoxy oligomer (EO) was CHS EPOXY 520 resin (Spolchemie, Ustin nad Labem, Czech Republic). The second type of epoxy oligomer was CHS EPOXY 520 resin, which was modified with 20 wt.% active diluent furfuryl glycidyl ether (FGE) (LLC “DOROS,” Yaroslavl, Russia) [53,54].

EO was modified with polysulfone (PSU) PSK-1 (JSC NIIPM, Moscow, Russia) with a molecular weight of 35,000 g/mol or polyethersulfone (PES) ULTRASON E-2010 (BASF, Florham Park, NJ, USA) with a molecular weight of 34,000 g/mol. The content of PSU or PES in EO is from 5 to 20 wt.% of the EO mass. EO with 20 wt.% FGE was modified with only 15 and 20 wt.% PSU based on the weight of EO [53].

Thermoplastics were dissolved in EO at a temperature of 100 °C and with constant stirring with a mechanical stirrer until complete combination. However, the addition of active diluent FGE to the epoxy polysulfone mixture is carried out at a temperature of 60–80 °C.

The obtained polymer mixtures were cured with hardeners amine type triethanolamine titanate (TEAT (JSC CHIMEX Limited, St. Petersburg, Russia)). The TEAT curing agent was introduced in an amount of 10 wt.% by weight EO or (EO + 20 wt.% FGE).

### 2.2. Matrices Samples

The preparation of polymer samples consisted in pouring the prepared mixtures into silicone molds. Then the binder was evacuated several times for 1 h at a temperature of 70–100 °C. Next, the compositions were cured for 8 h at 160 °C, after which the resulting bars were processed on a grinding machine. Thus, bars for bending tests under conditions of low-speed impact with a size of 5 mm × 5 mm × 40 mm were obtained.

### 2.3. Samples GFRP

Unidirectional GFRP was obtained by filament winding. In this case, RVMPN 10-400 glass fibers (NPO Stekloplastik, Andreyevka, Russia) were used. According to the manufacturer, the roving, made from a magnesia-aluminosilicate glass, consists of monofilaments with a diameter of 10 microns, a tensile strength ~2.3 GPa, elastic modulus ~75 GPa, and the sizing is for epoxy binders. It is well known that the introduction of thermoplastic modifiers into epoxy resins significantly (by several orders) increases the viscosity of the binder [43,51,55,56]. The increase in viscosity is the main obstacle to obtaining composites with high fiber content and low porosity. For the manufacture of samples of reinforced plastics, a technological scheme developed for the impregnation and tension of fibers using high-viscosity binders was used. The sample preparation technology is described in detail in [28]. Composites based on epoxy resin were cured for 8 h at 160 °C.

All the samples had the same technological and thermal background. The calculation of the amount of reinforced plastic components showed that the fiber content is practically independent of the composition of the binder (50–60 vol.%), and the porosity does not exceed 5 vol.%. The porosity was determined according to the standard [57]. Then, the wound rings were cut into 40 mm long segments. The section of the segments is 5 mm × 6 mm. The ratio l/h for the samples was ~6, and the distance between the supports was 32 mm [58].

Experiments under conditions of low-speed shock loading were carried out on a measuring complex developed at the FRC CP RAS (Figure 1). The loading unit (spring hammer) KPS-2 (Department of Experimental Technology, Federal Research Center for Chemical Physics, Russian Academy of Sciences, Moscow, Russia) is a modification of the falling load method, in which the speed of the impactor is controlled by a spring drive.

The impact driver has five spring charging stages, which allow measurements in the speed range from 1.2 to 7.0 m/s. We used loading speeds of 4.0 m/s and an impact energy of −13.0 J for polymer matrices and GFRP. 

The sample holding unit allows testing with three-point bending. This instrument assembly consists of a sample centering and holding system, and a strain gauge dynamometer, in which four strain gauges are combined into a complete Winston bridge. The distance between the supports of the tensometric dynamometer is fixed and equal to 32 mm. The signal from the strain gauge bridge is processed by a UTS-2M broadband amplifier (Department of Experimental Technology, Federal Research Center for Chemical Physics, Russian Academy of Sciences, Moscow, Russia), and then fed to a PCS-500 digital oscilloscope (Velleman, Legen Heirweg, Belgium) in the form of a loading diagram in the voltage-time coordinates. The photosensor generates an oscilloscope sweep trigger signal, which also passes through the UTS-2M amplifier. After testing, the shock-loading oscillogram is stored in the RAM of the computer. Subsequent signal processing and calculation of strength and energy characteristics are carried out using a Microsoft Excel macro.

The following were calculated from the oscillograms:Bending strength for non-reinforced polymers (*σ_b_*) [59]:
(1)$$\sigma_{b}=\frac{3Pl}{2bh^{2}},$$
where *P* is the maximum load on the loading diagram, *l*—is the length of the working part of the sample (the distance between the supports), and *b* and *h* are the width and height of the sample, respectively.

2.Shear strength for reinforced plastics (*τ*) [58]:(2)$$\tau =\frac{3P}{4bh},$$
where *P* is the maximum load on the loading diagram, and *b* and *h* are the width and height of the sample, respectively.

3.The total energy of the material fracture:(3)$$E_{F}=\left[v-\frac{1}{2m}\int_{t_{0}}^{t_{i}}P\left(t\right)dt\right]\cdot \int_{t_{0}}^{t_{i}}P\left(t\right)dt,$$
where *v* is the speed of the hammer at the moment of contact with the sample, *m* is the mass of the hammer, and the integral expression is the area under the load-time curve from the initial moment of loading *t*_0_ to the moment of time *t_i_* [60].

For each type of matrix, tests were performed on 10–12 samples. Following that, the forms of fracture for each sample were assessed visually and according to the shock loading oscillogram (Figure 2). The results of this study showed that the samples have two types of fracture upon impact. In the first type, the sample is destroyed without any fragments, while in the second type, multiple cracking of the sample is observed with the formation of fragments. The first type of fracture is characterized by an oscillogram with a maximum load that is two times less than the second type of fracture. Thus, the results obtained for samples of the first type of fracture were rejected and not accepted in further calculations. After such rejection, about 5 samples remained in each batch of matrices.

A total of 5 samples were analyzed for each batch of fiberglass.

The obtained values were used to calculate the mean value, the standard deviation, and the limits of the confidence interval. The morphology of the fracture surface after impacting the samples was examined using a Phenom ProX scanning electron microscope (Thermo Fisher Scientific, Waltham, MA, USA). The scheme of microscopic analysis of surface fracture for polymer matrices and fiberglass based on them is shown in Figure 3.

## 3. Results and Discussion

### 3.1. Matrix Impact Resistance

Loading diagrams for modified and unmodified epoxy matrices during bending under low-velocity impact conditions are shown in Figure 4. It can be seen that the samples are destroyed almost “in a flash” when the limit load is reached. The number of thermoplastic modifiers introduced does not affect the appearance of the “σ-t” diagram. An extremely short loading time of the sample is observed. Approximately 6.6–0.7 µs elapses from the moment the load is applied to the complete fracture of the samples. However, the height of the peaks depends on the composition of the epoxy matrices. From the diagrams obtained, the bending strength and the total energy of fracture were calculated.

Figure 5 shows bending strength and fracture energy for PSU- or PES-modified epoxy matrices. Figure 5a (curve 1) shows how the strength σ and total fracture energy E_m_ of epoxy polysulfone matrices change during bending under low-velocity impact conditions. The introduction of 5–10 wt.% PSU shows a trend towards a decrease in strength. In this case, the values of σ decrease by 10% compared to the unmodified matrix (from 164 MPa to 146 MPa). At a content of more than 10 wt.% PSU in the epoxy polymer, the strength σ begins to increase. The bending strength reaches its maximum for the EO + 20 wt.% PSU matrix and is equal to 200 MPa. A similar trend is also noticeable for the E_m_ energy: in the concentration range from 5 to 10 wt.%, the fracture energy decreases by 30% (from 32 kJ/m^2^ to 23 kJ/m^2^); at a polysulfone concentration of more than 15 wt.%, the increase in E_m_ is 20% (from 32 kJ/m^2^ to 39 kJ/m^2^) (Figure 5b, curve 1).

The introduction of 15 to 20 wt.% PSU into the EO + 20 wt.% FGE epoxy matrix does not increase the impact strength of the matrices (Figure 5a, “x” points) and their fracture energy (Figure 5b, “x” points). The values of strength σ and fracture energy E_m_ are at the level of the unmodified matrix (σ ≈ 150 MPa, E_m_ ≈ 28 MPa).

The introduction of PES in the concentration range of 0–15 wt.% into the EO matrix leads to a decrease in strength (Figure 5a, curve 2) and their fracture energy (Figure 5b, curve 2). Bending strength under low-velocity impact conditions decreases from 160 MPa to ~126 MPa, and fracture energy from 32 kJ/m^2^ to ~20 kJ/m^2^. At a PES concentration of 20 wt.% in EO, the value of σ decreases to 75 MPa and the value of E_m_ to 10 kJ/m^2^. Considering that PSU and PES belong to the same class of heat-resistant, rigid-chain polymers, the reduction in strength and fracture energy of epoxy polyethersulfone matrices seems unexpected.

The change in the strength of the modified matrices, in this case, is apparently associated with their structure and failure mechanisms.

Figure 6 shows photomicrographs of the surface of an unmodified epoxy matrix after low-velocity impact fracture.

Figure 6a corresponds to the lower part of the sample experiencing tensile stresses at the moment of impact. The micrograph shows the areas of origin and propagation of primary microcracks. Such areas are characteristic of the lower half of the fracture surface. The morphology of these areas is shown in more detail in Figure 6b,c. In Figure 6b, apparently, a microcrack occurs even before the moment when the ultimate loads are reached, which determines the strength of the matrix. Then the microcrack propagates in radial directions from the nucleation center (1) and gradually slows down towards the edges of the region (2). The deceleration of the microcrack is seen from the developed surface topography (limited by lines 2 and 3). In Figure 6c, one can see a similar region of primary crack initiation. The difference from Figure 6b in this case is that this area was formed not from an edge crack, but from a defect or local rupture of the matrix (1). In the upper half of the fracture surface of the matrix (Figure 6c), there is a surface formed during the splitting of two-cantilever beams. When splitting, normal stresses act. Apparently, in the case of beam bending in this part of the sample, fracture also occurs from normal stresses. The developed surface of the fracture indicates a lower crack propagation rate than in the lower half of the sample.

Analyzing the micrographs presented in Figure 6, it can be assumed that under the three-point loading of the sample, its fracture occurs in several stages (Figure 7). In the first stage, microcrack initiation centers are formed (Figure 7a) in the lower half of the sample, which experiences tensile stresses. In this case, the centers of occurrence of microcracks are fairly evenly distributed over the cross-section. Such centers can arise either from an edge crack or form in the bulk of the sample. In the second stage (Figure 7b), microcracks begin to propagate in radial directions from the center of their origin. As the microcrack front advances, its propagation velocity decreases. Then (Figure 7c) microcracks merge into a common network of macrocracks, covering the entire cross-section of the sample. When the network of macrocracks reaches half of the cross-section of the sample, the beam is destroyed from normal stresses by the splitting mechanism. The rate of crack propagation slows down even more. Further advancement of the macrocrack to the edge of the sample leads to the loss of its integrity. Generalizing the stages of initiation and propagation of cracks under the shock loading of a beam, it can be assumed that the initial stage of the fracture determines the impact resistance of the material, i.e., the resistance of the matrix structure to the formation and propagation of microcracks.

Figure 8 shows micrographs of the lower half of the surface fracture of epoxy matrices containing 20 wt.% PSU or PES. EO + 20 wt.%PSU composition matrices have a matrix-dispersion structure (Figure 8a). However, at such a high thermoplastic concentration, inverted phases are formed: the matrix is enriched in PSU (I), and the dispersed phase is in EO (II) [37]. The micrograph shows areas of microcrack formation (1) in the same way as for unmodified matrices. In this case, a microcrack forms in the EO(II) rich phase.

Further, the fracture occurs according to the fracture model described above. The high dissipative capacity of the PSU-enriched matrix at the initial stage of fracture is due to its microplasticity, which ensures high strength and fracture energy upon impact (see Figure 5, curve 1). The presence of the active diluent FGE in the composition of the epoxy matrix shifts the formation of the inverted phase structure to the region of high PSU concentrations [37]. In this case, at a PSU concentration of 20 wt.%, a matrix with interpenetrating structures is formed (Figure 8b). Structure 1 consists of a matrix enriched in EO (I) and a dispersed phase enriched in PSU (II). Structure 2 is an inverted structure 1: the matrix is enriched with PSU(II), and the dispersion is EO (I). The micrograph shows the places of formation and propagation of microcracks (circled by dotted lines). The initiation of microfracture occurs in structure 2, which has a lower dissipative capacity. Taking into account the above assumption about the matrix fracture mechanism, it can be assumed that interpenetrating structures dissipate the impact energy to a lesser extent. This conclusion is confirmed by lower values of strength and fracture energy of such matrices upon impact (Figure 5, points “x”).

The epoxy matrix was modified with 20 wt.% PES also does not achieve full phase inversion (Figure 8c,d), as well as for the matrix of composition EO + 20 wt.% (FGE + PSU). During curing, a “matrix-dispersion” type structure was formed. Dispersion (1) is enriched in epoxy oligomer, and matrix (2) has an inverted structure: continuous phase (I) is enriched in PES, and the dispersed phase is in EO. An important point is that during the fracture of the matrix, EO + 20 wt.% PES, cracks propagate mainly along the interface between structure (1) and structure (2) (Figure 8c). Such adhesive fracture is observed both in the lower half of the fracture surface and in the upper one. Moreover, in the lower half of the surface, microcrack initiation occurs in structure (1), the matrix of which is enriched in EO (Figure 8d). The fracture of this region is cohesive. As noted above, the EO + 20 wt.% PES matrix is characterized by the lowest values of strength and fracture energy (see Figure 5). Low adhesion between the phases did not allow reaching the values of σ and E_m_, which are typical for the EO + 20 wt.% FGE + 20 wt.% PSU matrix, which has a similar structure.

### 3.2. Impact Resistance of GFRP

Figure 9 shows a sample of GFRP with typical degradation after impact. All GFRP specimens are stratified approximately in the median plane, regardless of the type of matrix and the amount of modifier introduced, which confirms the fracture of the composite from the action of tangent stresses. A significant area of damage is also visible at the impact site in the center of the sample and at the supports on the bottom of the sample, about 5 mm from the end.

Figure 10 shows typical “force F vs. time t” loading patterns recorded during the impact of GFRP winding samples based on unmodified and modified epoxy matrices. For all the studied composites, the nature of the change in load from time to time under loading until the moment of the first delamination and the formation of the first group of cracks, which determine the strength of the material, practically does not change and does not depend on the composition of the epoxy matrices. This moment corresponds to the first peak in the diagram. In most cases, the load changes almost linearly up to the maximum value, and its growth also does not depend on the amount of modifier.

A slight non-linearity is present near the top of the first peak. A more detailed analysis of the extent of this non-linearity showed that if the loading time, until the first crack appears, is taken as 100%, then the loading time in the nonlinear section is several percent. Due to the fact that the length of the nonlinear sections is small, it can be assumed that the deformation of glass-reinforced plastics occurs quasi-elastically until the moment of growth of the first crack. The rate of stress growth in the sample during its loading remained constant.

The height of the peaks describes the strength of the material and depends on the composition of the epoxy matrices, as well as the type and amount of thermoplastic modifier. For GFRP based on epoxy matrices containing PSU, the height of the peaks is noticeably higher than for the original GFRP. GFRP, whose matrix contains 20 wt.% PES, perceives an even smaller load upon impact.

The subsequent fracture of GFRP is accompanied by multiple cracking. In this case, after the first peak, which determines the strength of the material, several peaks are observed in the diagrams, the height of which decreases as the samples are destroyed. Their height depends on the composition of the epoxy matrices. So, for epoxy fiberglass (see Figure 10, curve 1), the occurrence of the first delamination leads to a decrease in the perceived load by approximately two times, which indicates significant damage to the sample. At the same time, for composites based on epoxy resin modified with polysulfone (Figure 10, curves 2 and 3), after reaching the ultimate strength, the load decreases less, which indicates less damage to the composites. For example, fiberglass-based on a matrix containing 20 wt.% PSU (Figure 10, curve 2) or 20 wt.% PES (Figure 10, curve 4) is capable of absorbing up to 90% of the maximum load during secondary cracking. For GFRP based on epoxy matrix EO + 20 wt.% FGE + 20 wt.% PSU (Figure 10, curve 3), the level of perceived loads after the first failure is slightly reduced (to approximately 75%).

From the received diagrams, the strength and total fracture energy of wound GFRPs under low-velocity impact conditions were calculated (Figure 11).

Figure 11a shows how the shear strengths of fibers reinforced plastics change under dynamic loading conditions. With the introduction of polysulfone into the epoxy matrix of fibers reinforced plastics, the shear strength practically does not change in the concentration range of 0–5 wt.% PSU (Figure 11a, curve 1). In this case, the values of τ change in the range of 72–77 MPa. An increase in the PSU concentration in the fiberglass matrix to 10 wt.% affects a noticeable increase in strength. The strength value τ = 82 MPa, which is 14% higher than the strength of the reference fiberglass. At a content of 15 wt.% PSU in EO, the shear strength of fiberglass increases to 87 MPa (by 20%). A further increase in the concentration of the thermoplastic leads to a slight decrease in strength τ (up to 80 MPa).

When the EO + 20 wt.% FGE epoxy matrix is modified with PSU (Figure 11a, x-points), high strength composites are not achieved, as is the case with EO + 20 wt.% PSU-based GFRP. The shear strength of fiberglass is 64 MPa at 15 wt.% PSU and 70 MPa at 20 wt.% PSU. This decrease in shear strength is probably related to the structure of the fiberglass matrix, the formation of which is influenced by the active thinner FGE.

Another relationship is observed when the epoxy fiberglass matrix is modified with polyethersulfone (Figure 11a, curve 2). The shear strength of GFRPs decreases with increasing PES concentration. The maximum decrease in τ values is observed at a PES concentration of 10 wt.%; the strength is 58 MPa. With an increase in the concentration of PES in EO to 15–20 wt.%, the shear strength of fiberglass increases to 66–69 MPa. Just as for GFRP based on the EO + 20 wt FGE + PSU matrix, the decrease in strength can be explained by the features of the formation and fracture of modified matrices.

Figure 11b shows how the total fracture energy of a fiberglass epoxy matrix modified with PSU or PES changes. For unmodified fiberglass epoxy matrix, the values of total fracture energy are registered in the range of 615–741 kJ/m^2^ [50].

For the fiberglass epoxy-polysulfone matrix (Figure 11b, curve 1), with an increase in the PSU thermoplastic concentration to 15 wt.%, the total fracture energy grows slightly, and the E values are in the range of 560–640 kJ/m^2^. A significant increase in the total fracture energy of GFRP occurs when 20 wt.% PSU is introduced into EO. The energy E_GP_ increases by 35% to a value of 832 kJ/m^2^. However, in contrast to the shear strength for GFRP based on EO + 20 wt FGE + PSU, the total fracture energy of such fiberglass remains at the level of composites with an EO + PSU matrix (see Figure 5b point “x”).

The fracture energy of GFRP based on EO + PES smoothly increases with an increase in the thermoplastic concentration up to 15 wt.% and reaches its maximum. The E_GP_ value at this concentration is 950 kJ/m^2^. The maximum increase in degradation energy of GFRP modified with PES is approximately 30% relative to the unmodified composite, the same as for epoxy-polysulfone fiberglass.

With an increase in the PES concentration to 20 wt.%, a slight decrease in the E_GP_ value to 870 kJ/m^2^ is observed. The change in shear strength and total fracture energy of GFRP under low-velocity impact conditions is associated with the structure of hybrid matrices, as in the case of modified matrices.

Figure 12 shows micrographs of the crack surface of a GFRP sample formed by shear fracture during the bending of a short beam. As a rule, when loading a short beam, a crack is formed in the central layers of the sample and propagates from its end to the center (impact site). Figure 12a shows that the crack rate is very high, and the matrix fracture surface (2) is practically smooth between the reinforcing fibers (1). It is possible to distinguish areas of the matrix with a more developed relief, which is typical for shear fracture. Thus, the delamination mechanism is adhesive-cohesive. The remains of the epoxy matrix are visible on the reinforcing fibers, which indicates good adhesion of the polymer matrix to the surface of the reinforcing fibers. In the micrograph (Figure 12b) of the central part of the sample, the surface relief caused by shear failure is much more developed. However, the fracture mechanism also remains adhesive-cohesive. Closer to the point of impact (the center of the sample), the crack slows down to such an extent that microplasticity begins to manifest itself more clearly in the polymer matrix. On the surface of the matrix, there are practically no areas with a smooth surface. Thus, the rate of crack growth during shear of a short beam under low-speed loading is not constant and slows down from the edge of the sample to its center.

Figure 13 shows the delamination surfaces of GFRP based on the EO+ PSU matrix after impact. It can be seen that the introduction of 20 wt.% PSU leads to the formation of interpenetrating structures. Structure 1 (see Figure 13) consists of an EO-enriched matrix. Structure 2 consists of a matrix enriched with PSU (I), and dispersion—EO (II). Large areas of inverted structure 3 are found on the entire surface of the delamination. The relief of the surface also depends on the speed of the crack. When the crack grows rapidly (Figure 13a), the surface topography is smoother than when the crack grows slowly (Figure 13b,c). It is important to note that the PSU(I) enriched phase of inverted structure 2 is more deformed at slow crack propagation (Figure 13c) than at high propagation velocity (Figure 13a). The high resistance of the EO + 20 wt.% PSU matrix to shear stress significantly increases the shear strength of GFRP and the fracture energy (see Figure 10, curve 1). The high crack resistance of reinforced plastics based on a matrix modified with polysulfone has been repeatedly noted in [50,51,52]. It was noted that at a sufficiently high concentration of PSU (15–20 wt.%) in EO, structures with an extended thermoplastic phase are formed during curing, and the greatest resistance of the matrix to crack growth is observed. First of all, the high crack resistance of epoxy polysulfone-reinforced plastics is due to the increased micro deformability of the heterogeneous matrix.

When using the FGE epoxy polysulfone matrix as a modifier, a heterogeneous matrix is also formed during the curing process (Figure 14). However, it is not possible to obtain interpenetrating phases in the GFRP matrix. Between the reinforcing fibers 1 (Figure 14a) in the epoxy matrix 2, a dispersed phase 3 enriched with PSU is visible. Along the length of the delamination surface, a PES-enriched phase is rarely encountered. Probably, in volumes constrained by reinforced fibers, a large amount of polysulfone remains in the epoxy matrix and does not form its own phase. The entire length of fiberglass delamination is characterized by a developed relief of the fracture surface, which changes little with a decrease in the crack velocity. However, the low content of the PES-enriched phase in the epoxy matrix does not lead to an increase in GFRP shear strength and fracture energy (see Figure 11).

Figure 15 shows micrographs of GFRP based on an EO + 20 wt.% PES matrix. During the curing of the mixed binder under conditions of limitation of free volume by reinforcing fibers, phase decomposition of the initially homogeneous system occurred, just as in the matrix. Due to the conditions of matrix formation, it is not possible to obtain a completely inverted system, as well as in the case of modification with polysulfone. Interpenetrating structures are observed. Structure 1 (see Figure 15) consists of a matrix enriched in EO and a dispersed phase enriched in PES. Structure 2 is an inverted structure 1: the matrix is enriched in PES and EO dispersion. In contrast to the unmodified system, the relief of structure 1 has a developed surface over the entire delamination area. This indicates a high resistance of this phase to crack growth. Structure 2 (inverted phase) is deformed to varying degrees, depending on the location on the crack surface. In the figure, due to the rapid growth of the crack, the PES-enriched continuous phase retains its original shape. In places where EO-dispersed particles have been extracted from PES, the shape of the walls of the thermoplastic phase remains almost spherical. With a decrease in the crack growth rate closer to the center of the crack surface (Figure 15b), the deformability of the inverted phase (2) manifests itself to a greater extent. The walls of the PES-rich phase are elongated along the direction of crack growth, which indicates shear failure closer to the impact site (Figure 15c); two structures are also observed; however, structure 2 in structure 1 is unevenly distributed. As mentioned above (see Figure 15 and its description), during the phase decomposition of a similar polymer system in a free volume, the adhesive interaction of interpenetrating structures in epoxy polyethersulfone matrices is low compared to epoxy polysulfone matrices. Apparently, the introduction of PES into the GFRP matrix does not lead to an increase in shear strength under low-velocity impact conditions (Figure 11a, curve 2). However, due to the high micro deformability of the inverted structures, the impact energy can be effectively dissipated (Figure 11b, curve 2).

### 3.3. Energy Correlation

Figure 16 shows the correlation between the fiberglass fracture energy E_gp_ and the matrix fracture energy Em at low-velocity impact. By increasing the energy content of the epoxy matrix with polysulfone, it is possible to increase the total fracture energy of GFRP based on EO + PSU (Figure 16, curve 1). After increasing E_m_ by 22%, the E_gp_ of fiberglass increases by 35%. The fracture energies for the EO + 20 wt.% FGE + PSU and GFRP matrices based on them (Figure 16, x points) fit well on the correlation curve for epoxy-polysulfone systems. Taking into account that the samples of composites were destroyed in the same way from shear stresses, it can be assumed that the fracture energy of reinforced plastics significantly depends on the fracture energy of matrices during bending under low-velocity impact conditions.

For epoxy-polyethersulfone systems, there is also a correlation between the fracture energies of matrices and GFRP based on them. However, in this case, with a decrease in the fracture energy of the matrices, the fracture energy of GFRP increases. This change in fracture energies can be explained that due to the weak adhesion of the EO-rich phases and the PES-rich phases (Figure 16); there is a decrease in the strength of the epoxy-polyethersulfone matrices and their fracture energy at low-speed impact. With a high degree of probability, an increase in the adhesive interaction of the phases will lead to a significant increase in the impact resistance of the EO + PES matrices.

## 4. Conclusions

The work is aimed at studying the impact resistance of epoxy oligomer matrices (EO) modified with polysulfone (PSU) or polyethersulfone (PES) and glass-reinforced plastics based on them under low-velocity impact conditions. The concentration dependences of the strength and fracture energy of modified matrices and glass-reinforced plastics are determined. It is shown that the type of the concentration curves depends on the type of the modifying polymer.

The strength σ and fracture energy E_M_ of thermoplastic-modified epoxy matrices change little in the concentration range from 0 to 15 wt.%; however, even with the introduction of 20 wt.% PSU into EO, the strength increases from 164 MPa to 200 MPa, and the fracture energy from 32 kJ/m^2^ to 39 kJ/m^2^. The opposite effect is observed for the EO + 20 wt.% PES matrix, where strength decreases from 164 MPa to 75 MPa and energy from 32 kJ/m^2^ to 10 kJ/m^2^.

A study of the morphology of modified matrices after fracture under low-velocity impact conditions showed that the formation and propagation of cracks occur in several stages. The greatest implementation of the strength of hybrid matrices occurs at the initial stage of fracture during the nucleation and merging of microcracks.

The effect of increasing the strength and fracture energy of modified matrices is also retained in GFRP. The maximum increase in shear strength (from 72 MPa to 87 MPa) is observed for GFRP based on the EO + 15 wt.% PSU matrix. For GFRP based on EO + 20 wt.% PES, the shear strength is reduced to 69 MPa. The reference value for the fracture energy of GFRP 615 is 741 kJ/m^2^. The GFRP fracture energy based on EO + 20 wt.% PSU reaches a maximum of 832 kJ/m^2^ and for GFRP based on EO + 20 wt.% PES–at 950 kJ/m^2^.

The structure of GFRP matrices formed in a volume constrained by reinforcing fibers differs significantly from the structure of unreinforced matrices. Extended structures of the thermoplastic phase in GFRP matrices are not formed over the entire volume of the material. However, the higher the extension of the thermoplastic structures, the more resistant the reinforced plastics are to impact action.

For GFRP based on EO + PSU and EO + PES, there is a correlation between the fracture energies of the matrices. For different thermoplastics, the dependences differ significantly and depend on the type of thermoplastic and the adhesive interaction of the phases formed during the curing of the hybrid binder.

The detailed data obtained as a result of this study will make it possible to refine existing and future approaches to predicting the life cycle of materials. This opens up new possibilities in the micromechanics of hybrid polymer matrices fracture and reinforced plastics under low-velocity impact conditions.

## Figures and Tables

**Figure 1 polymers-15-02958-f001:**
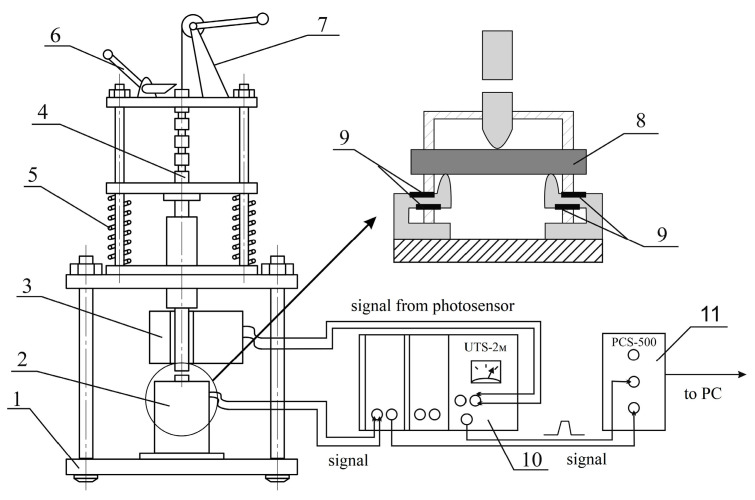
Scheme of the KPS-2 spring hammer device with a dynamometer for 3-point loading: 1—frame, 2—sample holding unit, 3—photo sensor, 4—hammer, 5—springs, 6—trigger, 7—spring tension device, 8—sample, 9—strain gauges, 10—UTS-2M amplifier, 11—PCS-500 storage oscilloscope.

**Figure 2 polymers-15-02958-f002:**
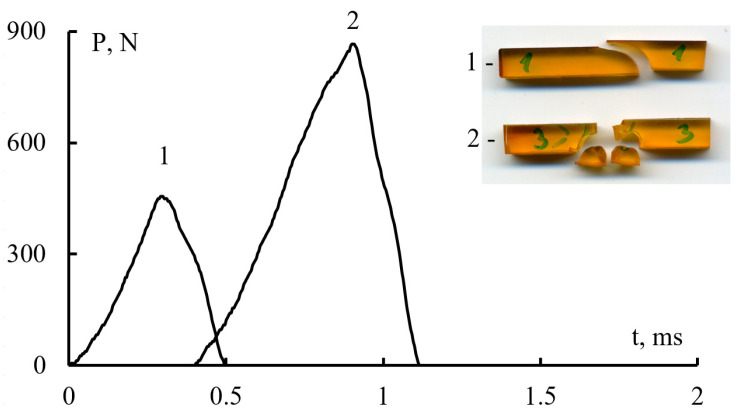
Oscillograms of loading in bending F-t and the view of samples of unreinforced matrices, fracture from a single crack (1) and as a result of multiple cracking (2).

**Figure 3 polymers-15-02958-f003:**
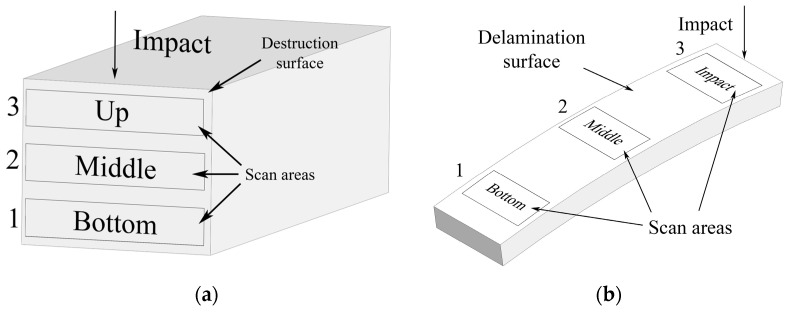
(**a**) Areas and procedures for microscopic analysis of fracture of matrix surfaces and (**b**) GFRP after fracture under low-velocity impact conditions.

**Figure 4 polymers-15-02958-f004:**
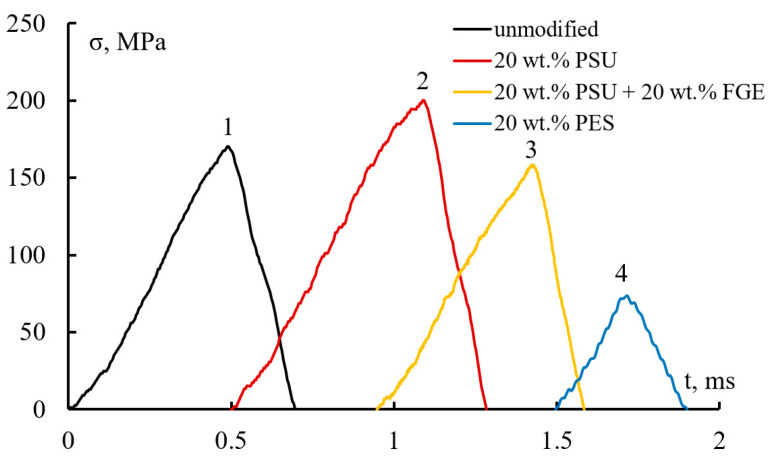
Loading diagrams of epoxy matrices during bending under low-velocity impact conditions.

**Figure 5 polymers-15-02958-f005:**
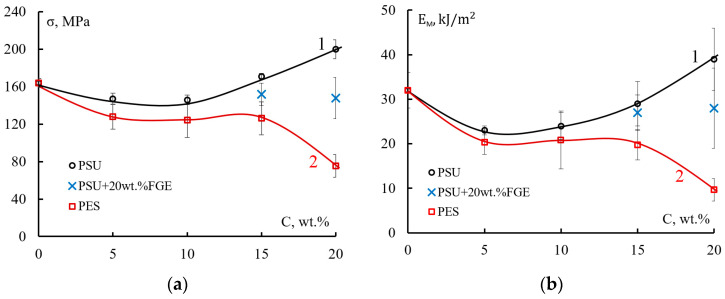
(**a**) Bending strength and (**b**) total energy of fracture of epoxy polysulfone (1), epoxy polysulfone modified with active diluent (2), and epoxy polyethersulfone (3) matrices.

**Figure 6 polymers-15-02958-f006:**
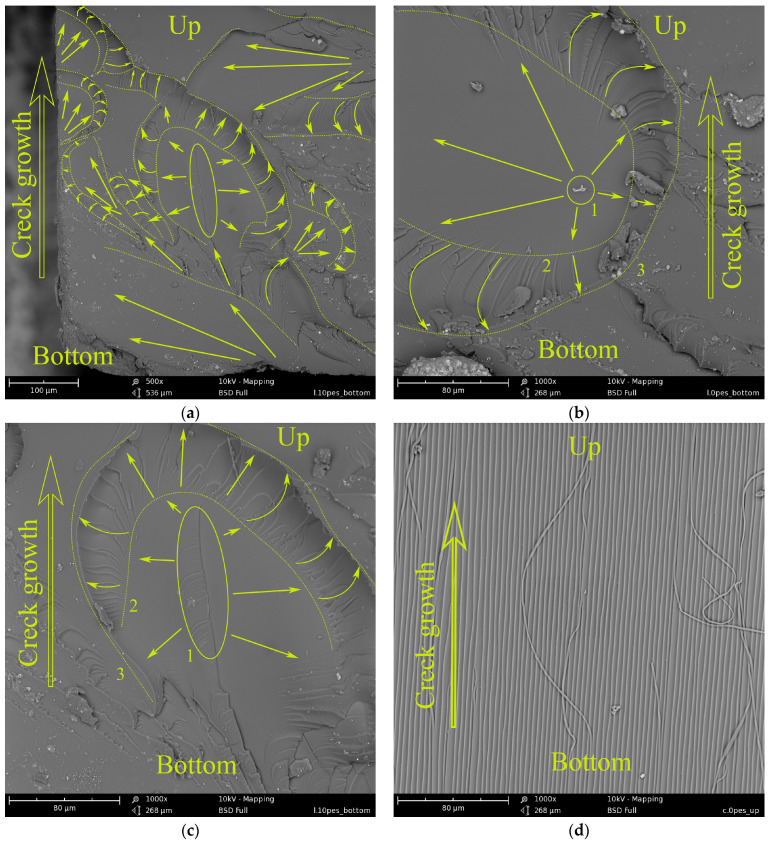
Surface fracture morphology of the unmodified (reference) epoxy matrix under three-point bending under low-velocity impact conditions: (**a**) lower part of surface fracture, magnification ×500; (**b**) the lower part of the fracture surface, a microcrack occurs in the bulk of the material, ×1000 magnification; (**c**) the lower part of the fracture surface, a microcrack originates from an edge defect, ×1000 magnification; and (**d**) upper part of the fracture surface, magnification ×1000.

**Figure 7 polymers-15-02958-f007:**
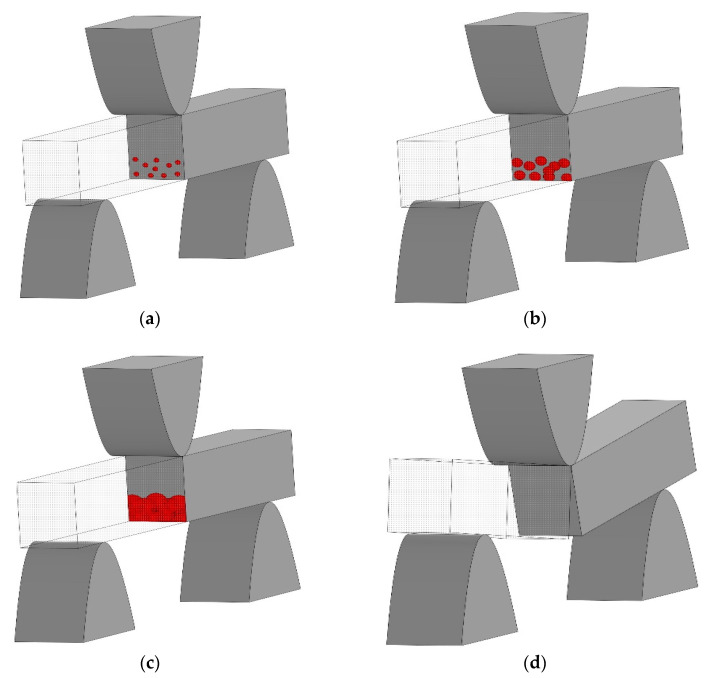
Scheme of fracture of a polymer beam under low-velocity impact: (**a**) initiation of fracture centers; (**b**) growth of microcracks from fracture centers; (**c**) merging of microcracks into a common crack; and (**d**) additional fracture of the material by the splitting mechanism.

**Figure 8 polymers-15-02958-f008:**
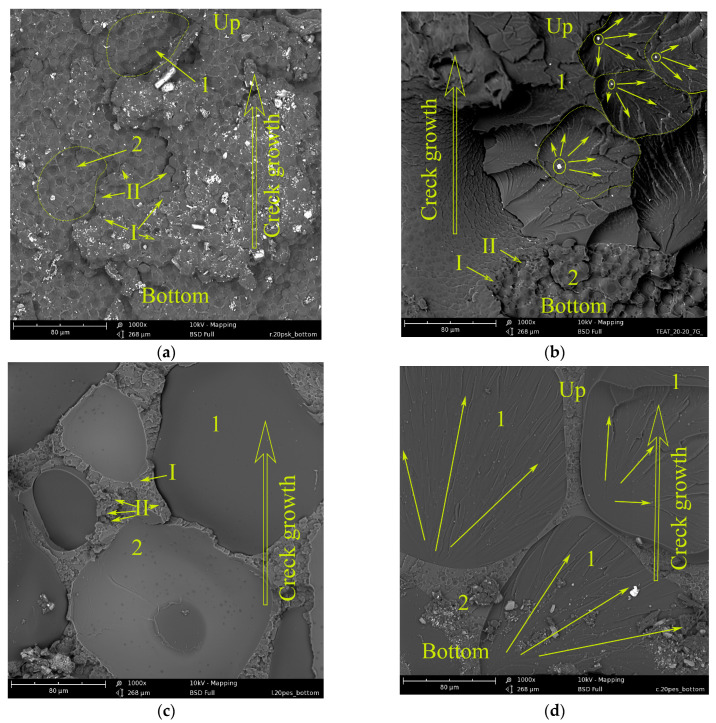
Surface morphology of fracture of modified epoxy matrices under three-point bending under low-velocity impact conditions: (**a**) EO + 20 wt.% PSU; (**b**) (EO + 20 wt.% FGE) + 20 wt.% PSU; (**c**) EO + 20 wt.% PES (adhesive degradation); and (**d**) EO + 20 wt.% PES (cohesive degradation).

**Figure 9 polymers-15-02958-f009:**
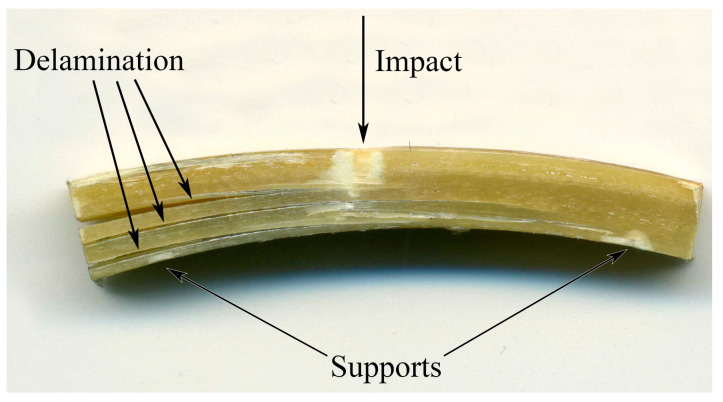
The standard view of a GFRP sample after a fracture.

**Figure 10 polymers-15-02958-f010:**
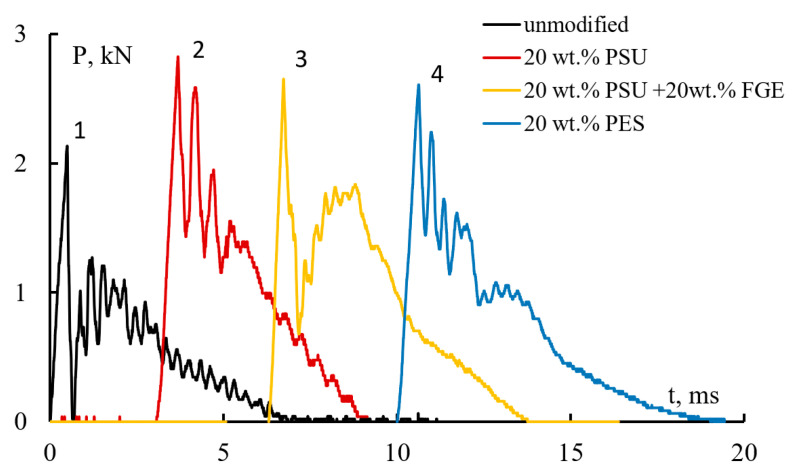
Typical degradation waveforms of GFRP based on unmodified and modified epoxy matrices: unmodified EO (1), EO + 20 wt.% PSU (2), EO + 20 wt.% FGE + 20 wt.% PSU (3), EO + 20 wt.% PES (4).

**Figure 11 polymers-15-02958-f011:**
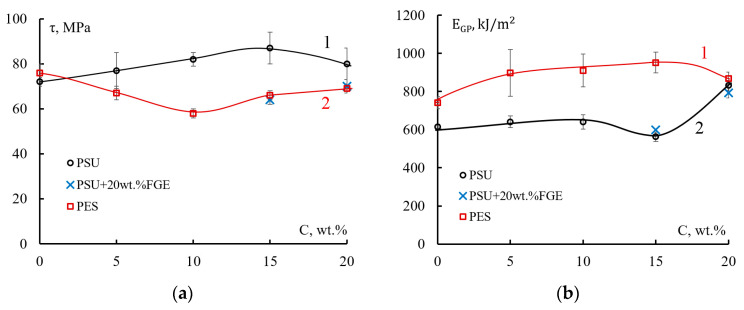
(**a**) Shear strength; (**b**) total fracture energy of winding glass fibers reinforced plastics based on epoxy polysulfone (1), epoxy polysulfone modified with an active diluent (x points), and epoxy polyether sulfone (2) matrices.

**Figure 12 polymers-15-02958-f012:**
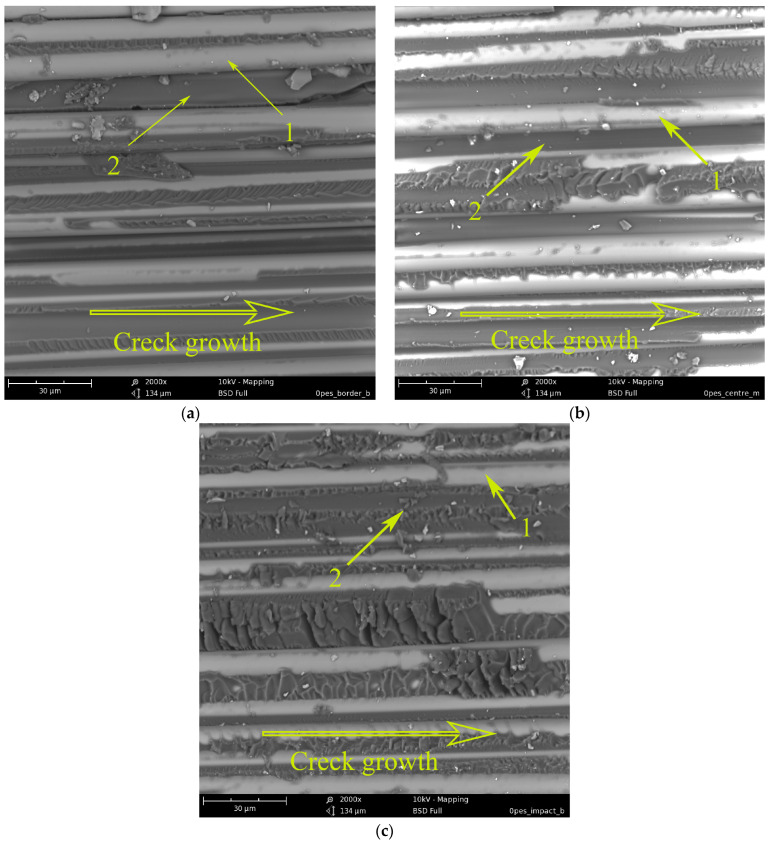
Morphology of the fracture surface of GFRP based on EO + TEAT (reference): (**a**) the edge of the fracture surface; (**b**) the middle of the fracture surface; and (**c**) a location near the impact.

**Figure 13 polymers-15-02958-f013:**
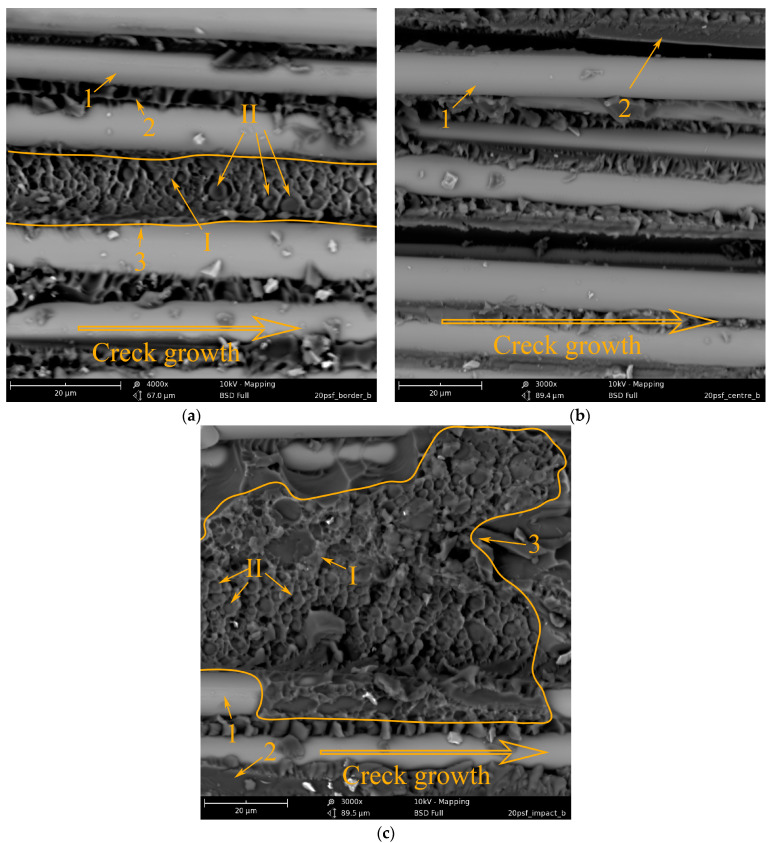
Morphology of fracture surface of GFRP based on EO + 20 wt.% PSU: (**a**) fracture surface edge; (**b**) the middle of the fracture surface; and (**c**) a location near the impact.

**Figure 14 polymers-15-02958-f014:**
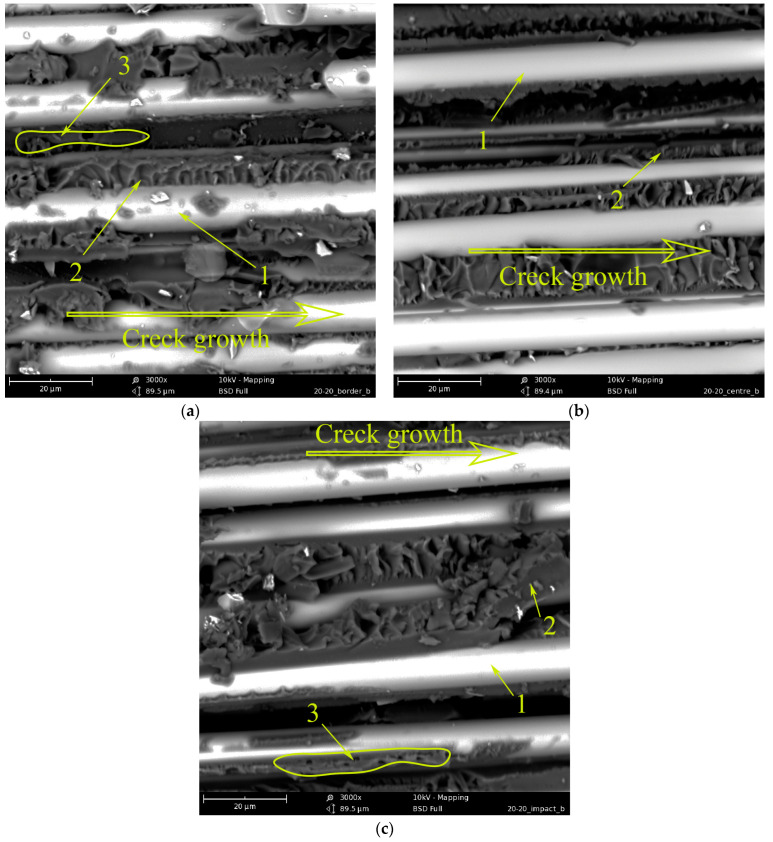
Morphology of fracture surface of GFRP based on EO + 20 wt.% FGE +PSU: (**a**) fracture surface edge; (**b**) the middle of the fracture surface; (**c**) location near the impact.

**Figure 15 polymers-15-02958-f015:**
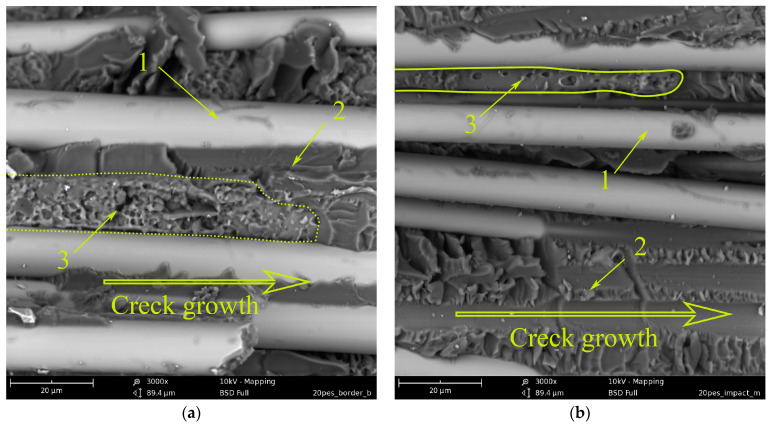
Morphology of fracture surface of GFRP based on EO + 20 wt.% PES: (**a**) fracture surface edge; (**b**) the middle of the fracture surface; and (**c**) location near the impact.

**Figure 16 polymers-15-02958-f016:**
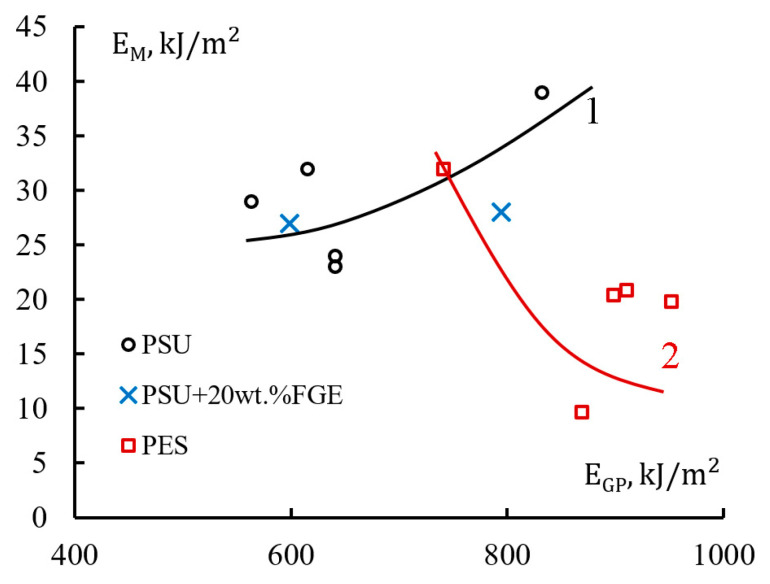
Correlation of total fracture energy of epoxy polymer matrices Em and GFRP based on them E_gp_.

## Data Availability

Not applicable.

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
