# Peer review of "Fracture of Epoxy Matrixes Modified with Thermo-Plastic Polymers and Winding Glass Fibers Reinforced Plastics on Their Base under Low-Velocity Impact Condition"

_polymers, 2023, doi:10.3390/polym15132958_

Round 1

Reviewer 1 Report

The paper reports an interesting research and can be published following the suggestions in annex.

The comments/suggestions are the following.

1 - In the title replace the word Destruction by Fracture

2 - In the texto, when appears destruction replace by fracture

3 - Page 1, line 33: where is "high strength" write "high specific strength"

4 - Page 2, line 43: where is "rein-forced" write "reinforced"

5 - Page 3, line 145: where is "size of 5x5x40 mm" write "size of 5mm x 5mm x 40 mm"

6 - Page 4, line 147: where is "obtained by winding" write "obtained by filament winding"

7 - Page 5, line 190: where is " Tensile strength in shear" write " Shear strength"

8 - Page 13, line 395: change the symbol related to strength

Line 408: write the figure number in Figure b

9 - Page 18, line 493: where "failureCloser" write "failure closer"

The title must be modified

The quality of English is acceptable, needing only minor editing

Author Response

Dear Reviewer!

We appreciate for the reviewer’s valuable comments and efforts towards improving our manuscript. We have revised the manuscript reflecting all the comments made. In response to the reviewer’s comments and suggestions, we carefully revised our manuscript, and in accordance with the reviewer request included additional data. Please find below a detailed point-by-point responses to all comments (reviewers’ comments in black, our replies in blue). We hope that the referee will find the revision adequately addressing the proposed suggestions. Please see attachment file.

Best regards, Gleb Yurkov

Reviewer 2 Report

The paper investigated the mechanisms of destruction of heterogeneous epoxpolysulfone and epox-ypolyethersulfone matrices and unidirectional glass-reinforced plastics based on them under low-velocity impact. The authors are encouraged to consider the following comments for necessary improvement.

1.      Abstract:

(1) A brief research background should be given to support the current research.

(2) The main research content should be given after the research background, and the authors should condense the relevant content.

(3) Such a description of “The mechanism of destruction of modified matrices in fiberglass differs from the mechanism of destruction of unreinforced heterogeneous systems.” should not appear in the abstract, and authors should give clear and concise results.

(4) Low-velocity impact should be added in abstract.

2.      Introduction:

(1) As authors said polymer composite materials are widely used aerospace, automotive, and civil engineering, because polymer composite materials are lightweight, higher mechanical and superior aging and fatigue resistance. Authors should enrich the references for the above research backgrounds. The following relevant studies can be reviewed to make necessary supplements in the research background, such as “Engineering Structures, 2023, 274: 115176.”, “Composite Structures. 2021, 261: 113285” and “Mechanics of Advanced Materials and Structures, 2023, 30(4):814-834.”.

(2) Shorten the introductory paragraph and merge similar content

(3) Please summarize the references and highlight the innovation of this paper in the last paragraph of the introduction.

3.      Materials and methods

(1) The mechanical performances of two type of epoxy should be provided.

(2) How the dispersed of thermoplastics in epoxy resin? Whether there are corresponding testing methods?

(3) The mechanical performances of glass fiber should be provided.

(4) The porosity does not exceed 5 vol % of all samples, specific characterization test means?

4.      Results and Discussion

(1) Give the mechanism explanations of decrease or increase of strength and total energy of fracture in Figure 5.

(2) The reference standard of impact testing should be given. Impact failure mode of GFRP sample is delamination, what kind of impact failure mode will be caused by increasing the span of the sample?

5.      Conclusions

The conclusions should be condensed, authors should point out the key results rather than repeating experimental rules.

It is Ok.

Author Response

(The authors gave the same response as above.)

Round 2

Reviewer 2 Report

The authors have responded well to the reviewers' comments.